# Hallucinogen Persisting Perception Disorder: Etiology, Clinical Features, and Therapeutic Perspectives

**DOI:** 10.3390/brainsci8030047

**Published:** 2018-03-16

**Authors:** Giovanni Martinotti, Rita Santacroce, Mauro Pettorruso, Chiara Montemitro, Maria Chiara Spano, Marco Lorusso, Massimo di Giannantonio, Arturo G. Lerner

**Affiliations:** 1Department of Neuroscience, Imaging and Clinical Sciences, University “G. d’Annunzio”, 66100 Chieti, Italy; giovanni.martinotti@gmail.com (G.M.); chiara.montemitro@gmail.com (C.M.); m.chiara.spano@gmail.com (M.C.S.); doloma2012@gmail.com (M.L.); digiannantonio@unich.it (M.d.G.); 2Department of Pharmacy, Pharmacology, Postgraduate Medicine, University of Hertfordshire, Herts AL10 9AB, UK; 3Institute of Psychiatry and Psychology, Catholic University of Sacred Heart, 00168 Rome, Italy; mauro.pettorruso@hotmail.it; 4Lev Hasharon Mental Health Medical Center, Pardessya 42100, Israel; alerner@lev-hasharon.co.il; 5Sackler School of Medicine, Tel Aviv University, Ramat Aviv 69121, Israel

**Keywords:** Hallucinogen Persisting Perception Disorder, flashbacks, hallucinogenic substances, LSD, psychedelics, visual disturbances, perceptual disturbances

## Abstract

Hallucinogen Persisting Perception Disorder (HPPD) is a rare, and therefore, poorly understood condition linked to hallucinogenic drugs consumption. The prevalence of this disorder is low; the condition is more often diagnosed in individuals with a history of previous psychological issues or substance misuse, but it can arise in anyone, even after a single exposure to triggering drugs. The aims of the present study are to review all the original studies about HPPD in order to evaluate the following: (1) the possible suggested etiologies; (2) the possible hallucinogens involved in HPPD induction; (3) the clinical features of both HPPD I and II; (4) the possible psychiatric comorbidities; and (5) the available and potential therapeutic strategies. We searched PubMed to identify original studies about psychedelics and Hallucinogen Persisting Perception Disorder (HPPD). Our research yielded a total of 45 papers, which have been analyzed and tabled to provide readers with the most updated and comprehensive literature review about the clinical features and treatment options for HPPD.

## 1. Introduction

Hallucinogens represent an enormous group of natural and synthetic agents [1,2]. The core features of hallucinogens include their being empathogenic and being able to induce alterations of consciousness, cognition, emotions, and perception. Their main characteristic is to profoundly affect a person’s inner processes and the perception of the surrounding world. The perceptual distortions are mainly visual, as in the vast majority of induced psychoses [3,4,5]. The hallucinogenic properties of many natural products were known for thousands of years: popular healers, “brujos”, and shamans used these substances in ancient times for medical, religious, spiritual, ritual, divination, and magical purposes. Nevertheless, the attention of western culture was drawn to psychedelics only at the beginning of the 20th century, but the turning point is considered to be 1938, the year in which the lysergic acid diethylamide, better known as LSD, was synthesized by Albert Hofmann. In the 1950s and 1960s, LSD was considered to have a therapeutic potential in the psychiatric field, allowing patients to access unconscious material in therapeutic settings. This has been recently re-evaluated with uncertain results. After a mass diffusion of hallucinogens in the 1960s and 1970s, current prevalence data [6] from the United States highlight that more than 180,000 Americans report a recent use of LSD, and 32,000 a recent use of phencyclidine. Nowadays, the intake of hallucinogens is associated with shamanic ceremonies, workshops of underground therapy and self-experiences. In these frameworks, hallucinogenic substances are most commonly used alone, while in rave parties and social events they are often part of a heavy polyvalent use that frequently includes Novel Psychoactive Substances. These compounds, easily available on the Internet without any cultural barrier and sometimes without any advice from the group of peers, have profoundly changed the drug scenario [7,8,9,10]. Their use is becoming widespread, also due to their low cost and appealing market strategies [11,12]. However, significant medical and psychiatric problems have been reported for subjects using these drugs [13], regardless of previous psychiatric antecedents [14].

This paper will focus on a rare, and therefore, poorly understood aspect of hallucinogen consumption: the total or partial recurrence of perceptual disturbances that appeared during previous hallucinogenic “trips” or intoxications and re-emerged without recent use [4,5]. These returning syndromes are defined “benign flashbacks”, or pervasive Hallucinogen Persisting Perception Disorder (HPPD). LSD is the model and prototype of the classical synthetic hallucinogen, and it is certainly the most explored and investigated substance associated with the etiology of this unique and captivating state [15]. HPPDs do not have a notable prevalence [16], and, therefore, they are frequently unrecognized [17,18].

Classifications used to delineate and outline persisting perceptual disorders are now clearer than in the past [18]. Two major subtypes of hallucinogenic substance-use related recurring perceptual disturbances have been identified and reported [18]: (1) HPPD I, also described and named as benign Flashback and Flashback Type; and (2) HPPD II, also named HPPD Type II [17,18]. HPPD I has a short-term, reversible and benign course. Although visual images may provoke unpleasant feelings, re-experiencing the first hallucinogen intoxication may not lead to significant concern, distress, and impairment in individual, familial, social, occupational, or other important areas of functioning [17,18]. The impairment is mild and the prognosis is usually good. Some of the patients do not report being annoyed by these phenomena: they may indeed consider them as “free trips” resembling psychedelic experiences without consuming a psychoactive substance. Contrarily and conversely, HPPD II has a long-term, irreversible or slowly reversible and pervasive course [17,19]. The impairment of HPPD II is severe and the prognosis is worse. Some of the patients fail to adapt and live with these long-lasting recurrent “trips”, and a consistent fraction needs to be constantly treated [19,20]. It has to be considered that the distinction between HPPD type I and HPPD type II has not yet been made in the Diagnostic and Statistical Manual of Mental Disorder, fifth edition (DSM-5) and it is still debated. HPPD type I is consistent with the diagnostic definition expressed by the International Classification of Disease, 10th (ICD-10), while HPPD type II better matches the DMS-5 criteria.

A vast list of psychoactive substances has been identified and linked with the development of this condition, including Magic Mushrooms (psilocybin) [21] and muscimol (*Amanita muscaria* (L.) Lam.) [22]; San Pedro cactus and Peyote (mescaline) [16,23]; ketamine [24]; dextromethorphan [25]; MDMA and MDA [26]; and cannabis and synthetic cannabinoids [27,28,29,30,31,32,33]. This condition has also been associated with the consumption of Ayahuasca, *Datura stramonium* L., *Salvia divinorum* Epling & Játiva, and *Tabernanthe iboga* (L.) Nutt., which contains ibogaine [17,18]. It is, therefore, clear that HPPD is not strictly associated with psychedelic consumption, but a number of hallucinogen-inducing substances may be correlated with its arising.

The aim of the present study is to review all the original studies about HPPD in order to evaluate (1) the possible suggested etiologies; (2) the possible hallucinogens involved in HPPD induction; (3) the clinical features of both HPPD I and II; (4) the possible psychiatric comorbidities; and (5) the available and potential therapeutic strategies.

## 2. Materials and Methods

We searched PubMed to identify original studies about psychedelics and Hallucinogen Persisting Perception Disorder (HPPD). The following search terms were used: “Hallucinogen Persisting Perception Disorder” OR “Hallucinogen Persisting Perceptual Disorder”. The search was conducted on 15 September 2017 and yielded 46 records. We included all original articles (open-label or double-blind trials, prospective or retrospective observational studies, and case reports) written in English. We included all studies describing perceptual distortions in patients with a previous history of substance consumption. Reviews, commentaries, letters to the editor, and studies enrolling adolescents were excluded. All the authors agreed on the inclusion and exclusion criteria. We excluded 17 records by reading the titles and abstracts. By reading the full texts of the 29 remaining articles, we found 25 papers meeting our inclusion/exclusion criteria, and we, therefore, included them in the qualitative synthesis (Figure 1).

## 3. Results

### 3.1. Suggested Etiologies 

HPPDs are poorly understood due to the enormous range and variability of recurrent sensory disturbances, and the multiple distinct subtypes [17,18].

The main neurobiological hypothesis is that LSD consumers might develop chronic disinhibition of visual processors and dysfunction in the function of the central nervous system (CNS) [4,34,35,36]. This disinhibition may be linked to an LSD-generated intense current [37] that may determine the destruction or dysfunction [18] of cortical serotonergic inhibitory interneurons with *gamma-*Aminobutyric acid (GABAergic) outputs, implicated in sensory filtering mechanisms of unnecessary stimuli [34,35,36,38]. The efficacy of some treatment options in a subject with HPPD, such as pre-synaptic α_2_ adrenergic agonists, selective serotonin reuptake inhibitor (SSRIs), benzodiazepines, and mood stabilizers would confirm this neurobiological hypothesis (see Section 3.2). Reverse tolerance or sensitization that emerges after LSD exposure may explain recurrent occurrences after the stimulus has been withdrawn [39]. Nonetheless, HPPD-like experiences, such as flashbacks, moments of derealization, and hyper-intense perceptions are reported in healthy populations and non-LSD exposed subjects [40]. Moving from biochemical receptor interactions towards macroscopic areas, a temporary or permanent impairment in the Lateral Geniculate Nucleus (LGN) has been hypothesized [4,41,42,43]. The LGN, which is located in the thalamus, is associated with visual perception pathways [41,42,43]. Recent research highlighted a brain dysfunction in patients with visual snow, located mainly in the right lingual gyrus [44], perhaps implying LSD involvement. Halpern et al. [40] suggested that HPPD can be due to a subtle over-activation of predominantly neural visual pathways that worsens anxiety in predisposed subjects after ingestion of arousal-altering drugs, including non-hallucinogenic substances. According to Holland and Passie, environmental triggering by specific situations or stimuli or other elements related to the original experience may be involved in flashback experiences [45].

### 3.2. Substances That Induce HPPD

Different substances have been associated with visual and perceptual disturbances (Table 1).

According to the literature, we found that the majority of HPPD cases have been induced by LSD or phencyclidine (PCP) (14 studies, 294 patients) [17,19,21,26,35,46,51,52,53,55,56,57,58,59].

The use of cannabis has been associated with the development of perceptual distortions in seven patients [29,46,48,49,61]. In one case, it was associated with 3,4-Methylenedioxymethamphetamine (MDMA) and in another case with PCP [48,49]. In two patients, visual distortion followed the consumption of synthetic cannabinoids [61].

Lauterbach et al. reported the unique case of HPPD induced by the atypical antipsychotic Risperidone [60].

### 3.3. Clinical Features

According to DSM-5, Hallucinogen Persisting Perception Disorder is the recurrence of perceptive disturbances that firstly develop during intoxication. The contents of the perception and visual imagery range extensively [17,19]. DSM-5 and previous DSM editions report a list of the most common symptoms experienced by HPPD patients, but only a few symptoms have been described in the professional literature. The main group of symptoms reported by Criterion A of the DSM-5 are visual disturbances. In fact, as in the vast majority of induced psychoses, visual hallucinations are notably more common than auditory [3]. Regardless, every perceptual symptom that was experienced during intoxication may re-occur following hallucinogen withdrawal. We report a list of the main literature-reported visual disturbances in Table 2.

A latent period may antecede the onset of returning visual occurrences. This latent period may last from minutes, hours, or days up to years, and re-emerge as either HPPD I or II with or without any recognized or perceived precipitator [17,19]. Episodes of HPPD I and II may appear spontaneously or they may be triggered by identified and non-identified precipitators [18]. Episodes may be continuous, intermittent, or sudden. With regards to this point, neither HPPD I nor HPPD II can really be considered as persisting in a narrow sense of the word. Additionally, their differential diagnosis can only be proposed in terms of prognosis rather than clinical presentation.

However, HPPD I usually onsets with warning “auras”, minor feelings of self-detachment, mild bewilderment, and mild depersonalization and derealization [17,18]. Conversely, the onset of HPPD II might be unexpected and abruptly detonate with bursting “auras”, deep feelings of self-detachment, acute depersonalization-derealization [19].

The frequency of recurrence of perceptual distortions is lower for HPPD I than HPPD II [18]. Prior substance users can voluntarily elicit or produce visual disturbances with or without known triggers [4,17,18]. After HPPD II onset, hallucinogenic events tend to occur more frequently, and their duration and intensity increase. Subjects might perceive a partial or total loss of control.

### 3.4. Mental Illnesses Comorbid with HPPD

Recent observations reported a co-occurrence with depressive [20] and anxiety traits [51] and severe mental illnesses such as Major Depressive Disorder [23], Bipolar Disorder [23,62], and Schizophrenia Spectrum Disorders [17,58]. However, HPPD I and HPPD II onsets are not necessarily accompanied by any prominent additional psychiatric disorder, thus representing an independent condition [17,18]. In particular, the onset of HPPD II is often linked to a clear negative mood and affect. Anxiety and depressive features might aggravate new episodes. Anxiety might also evolve into a panic attack. Anticipatory anxiety may antecede future visual aberration events, and avoidant behavior may limit and restrict regular normal functioning [17,18]. Recently, in a study carried out by Halpern, a comprehensive survey of 20 subjects reporting Type-2 HPPD-like symptoms was presented and evaluated. The dissociative symptoms were consistently associated with HPPD, suggesting that HPPD is in most cases due to a subtle over-activation of predominantly neural visual pathways that worsens anxiety in predisposed subjects after the ingestion of arousal-altering drugs, including non-hallucinogenic substances. The authors report that many perceptual symptoms reported were not first experienced while intoxicated, and are partially associated with pre-existing psychiatric comorbidity, tempering the direct role of hallucinogens in the etiopathology of the disorder [40].

Only two observational studies and one case report evaluated psychotic patients with comorbid HPPD [57,58,60] (Table 3). Two observational, cross-sectional studies compared schizophrenic patients with prior use of LSD who developed HPPD (SCZ+HPPD, 49 patients) with those who did not (SCZ, 57 patients), for a total of 106 patients [57,58]. No differences between the two groups have been found with respect to demographic characteristics, age of psychotic onset, age of drug use onset, and type of substances abused [57,58]. As expected, SCZ+HPPD patients reported more distressing and horrific LSD experience (“bad trips”) (*p* < 0.05) [57]. Interestingly, the positive subscale of the Positive and Negative Syndrome Scale (PANSS) did not differ between the two groups. On the contrary, SCZ+HPPD patients showed lower scores in the PANSS negative subscale, the PANSS General Psychopathology Subscale, and the PANSS total scores (*p* < 0.05) [57]. Moreover, 67% of the schizophrenic patients comorbid with HPPD were able to distinguish between perceptual distortion and psychotic hallucinations [58], and 9 out of 12 patients could identify precursory cues for perceptual distortion (substance-induced cues, situational cues, and mental cues) [58]. Lauterbach et al. [60] reported a case of HPPD comorbid with psychosis, in which visual distortions were induced by antipsychotic treatment. Interestingly, the patient did not report any history of previous substance abuse [60]. The patient was treated with Risperidone, Clonazepam and Trazodone, and she reported visual disturbances resembling HPPD, in particular, illusions, after three subsequent Risperidone dosage increases [60].

### 3.5. First-Line Medications

Pre-synaptic α_2_ adrenergic agonists are a treatment with a low side-effect profile for patients with a previous history of substance-related disorders. Symptoms alleviation has been reported in some patients treated with these drugs [17,18,52,63]. The effectiveness may be based on the evidence that clonidine may elevate plasma GABA levels in humans, having a benzodiazepine-like effect. Clonidine may also decrease locus coeruleus activity, leading to a reduction of adrenergic activity [64], which can be effective in the management of PTSD [65]. Therefore, as in PTSD-related recurring flashbacks, some visual disturbances could be associated with excessive sympathetic nervous activity. Thus, these visual distortions could be ameliorated by Clonidine [52,63].

A dosage of 0.75 mg/die of Clonidine has been evaluated as a treatment option for nine HPPD patients [51,59] (Table 4). The total remission has been reported in a single patient with flashbacks and anxiety treated with 0.25 mg of Clonidine three times a day for two months [59]. In the 2 months open study on eight HPPD patients, the Clinical Global Impression (CGI) and Patient’s Severity Perception significantly decreased between entry and endpoint scores [51], although two patients dropped out at week 3 and week 5, respectively [52]. Lofexidine (0.2–0.8 mg/day) is a sympatholytic centrally acting α_2_ presynaptic adrenergic agonist that showed similar efficacy in some cases [23,65,66].

Benzodiazepines may be useful and effective in eliminating benign HPPD I and ameliorating, but not completely eradicating, pervasive HPPD II symptoms [18,67]. The effectiveness of Benzodiazepines may be related to their activity on the cortical serotonergic-inhibitory inter-neurons with GABAergic outputs [2,4]. Alprazolam (0.25–0.75 mg/day) has been prescribed with some success and Clonazepam (0.5–1.5 mg/day) appears to be the most reliable and effective benzodiazepine even at low doses [17,18,51,67]. Higher doses (4 mg/day) have also been used with good outcomes [68]. Clonazepam may act on serotonergic systems, improving, enhancing, and augmenting transmission [17,18,51,67], thus promoting alleviation and a marked improvement [51,67]. Clonazepam has been evaluated in three case reports and one open-label trial by Lerner [19,50,51]. In the clinical trial, 16 HPPD patients were treated with a Clonazepam dosage of 2 mg/day [51]. Their symptoms improved significantly after treatment initiation and the improvement persisted during a 6-month follow-up after treatment discontinuation [51]. The same author reported two cases of cannabis-induced visual disturbances and correlated anxiety features. In both cases, Clonazepam (2 mg/day) was effective in improving symptoms, but focal visual disturbances without anxiety (trailing phenomena in one case, and black moving spots in the second case) persisted during and after therapy [19]. More recently, Clonazepam (6 mg/day) has been proved to be effective in improving cannabis-induced HPPD symptoms [50]. On the other hand, the intrinsic abuse potential of benzodiazepines might be inconvenient in certain individuals with a past history of substance use [17,18]. Given the benign nature of HPPD I, the use of benzodiazepines should be proposed only for severe cases, in the acute phase, and for the short term.

HPPD patients appear to be sensitive to first-generation antipsychotics at low doses, requiring monitoring of extrapyramidal side effects. Haloperidol [69] and Trifluoperazine [70] were reported to be helpful. Perphenazine (4–8 mg/day) [17,23], Sulpiride (50–100 mg/day) [23], and Zuclopenthixol (2–10 mg/day) [17,23], at very low doses, are well tolerated and may be an effective treatment. Some of the long-acting first-generation antipsychotics may still be useful in co-occurring Psychotic Spectrum Disorders and HPPD II [58]. In one study, haloperidol was noted to reduce hallucinations, but an exacerbation of flashbacks in the early phases of treatment was highlighted as well [1,69].

The use of second-generation antipsychotics in HPPD patients without comorbid psychotic disorders is debated. Anderson recently reported the case of a young woman presenting prolonged and distressing multimodal pseudo-hallucinations, depressive symptoms, and anxiety, who was treated with Risperidone for three months without any significant improvement [48]. At the same time, conflicting evidence exists on the antipsychotics effects in psychotic HPPD patients. One study did not report differences in antipsychotic treatment response between SCZ and SCZ+HPPD patients [58]. On the other hand, a more recent study has shown the ineffectiveness of antipsychotic medications in an SCZ+HPPD population [57].

Risperidone was usually prescribed due to its proven efficiency in the treatment of perceptual disturbances in Psychotic Spectrum Disorders, mainly in Schizophrenic Disorders. LSD seems to work as a partial agonist of postsynaptic serotonin receptors. Therefore, Risperidone, which is a strong antagonist of both postsynaptic 5-HT_2_ and D_2_ receptors, was expected to be convenient. In contrast with this supposition, Risperidone at recommended [71] and lower doses [72] worsens visual disturbances and accompanying anxiety, or does not show any effect [54]. This was presumably due to Risperidone’s α_2_ presynaptic antagonism and noradrenaline release [34]. In addition, Risperidone was associated to the re-experiencing of visual disturbances in some patients suffering from schizophrenia with a past history of LSD use [73]. One psychotic patient treated with Risperidone, Clonazepam, and Trazodone reported visual disturbances resembling HPPD after three subsequent Risperidone dosage increases [60]. At the same time, Risperidone has been shown to be effective in improving PCP-induced HPPD with anxiety in one patient, while in the same patient Olanzapine produced symptoms exacerbation [21].

Evidence not included in our systematic review suggested that low dosages of atypical antipsychotics may be useful, specifically Aripiprazole (5–10 mg/day) [23], also because of its efficacy in substance and alcohol use disorders [74].

Visual oddities and disturbances with sudden paroxysmal onset have been interpreted as visual seizures and prompted the use of antiepileptic drugs in HPPD. This consideration helped to explicate the efficacy of benzodiazepines and led to the prescription of Phenytoin [75,76]. Today, Phenytoin is not used for HPPD treatment due to its troubled side effect profile. Medications such as Valproic Acid (200–600 mg/day), Carbamazepine (200–600 mg/day), Oxcarbamazepine (300 mg/day), Gabapentin (300–900 mg/day), Topiramate (25–100 mg/day), and Lamotrigine (50–100) may be useful [23], also because of their efficacy in substance and alcohol use disorders [77,78,79]. In a single case of HPPD symptoms and electroencephalographic (EEG) abnormalities, compatible with toxic encephalopathy, the visual hallucinations that recurred at any alcohol ingestion improved, but did not disappear with the use of Valproic Acid (1500 mg/day) [46]. Levetiracetam has shown to reduce some visual symptoms as well as HPPD related-depersonalization and derealization [80]. Lamotrigine has shown to be efficacious in a recent severe case of HPPD with some EEG abnormalities (Anderson et al., 2018). These medications may also be helpful when visual disturbances are accompanied by co-occurring mood swings and mood disorders.

Antidepressant medications could help in the management of co-occurring HPPD II with anxiety and depressive disorders [17,18,20,51,67]. HPPD II alone does not appear to be an appropriate target. There are questionable and controversial results regarding Sertraline, which has been reported to worsen [81] as well as to improve visual disturbances. Amelioration following long-term administration of SSRIs was attributed to the down-regulation of 5-HT_2_ receptors, providing more evidence to corroborate the serotonergic mechanisms underlying this condition. Other prescribed SSRIs did not show any benefits. Norepinephrine reuptake inhibitors (NRIs) such as Reboxetine have been tried with some success in LSD-induced HPPD symptoms comorbid with Major Depressive Disorder [20]. Agomelatine, given its peculiar function on neurotrophic factors [74], could have some benefits on the syndrome, although no data are available until now.

### 3.6. Second Line Medications 

Naltrexone has been usually used, alone or with other medications, in chronic patients with continuous visual imagery that previously did not respond to other medications [17,18].

Calcium Channel Blockers and Beta Blockers may be helpful in patients with co-occurring HPPD II and anxiety disorders [18]. Propanolol at low (20–60 mg/day) and high doses (240 mg/day), as well as Atenolol 25–50 mg/day, have been used to diminish accompanying anxiety of visual imagery [18,23]. Investigations of HPPD patients with EEG mapping showed that HPPD is represented by disinhibition [35] in the cerebral cortex [34]. The rationale behind this interesting and novel approach is that improving sensory gating by dopaminergic enhancers may cause an inhibition of catechol-*O*-methyl transferase (COMT), that may improve HPPD symptomatology.

### 3.7. Brain Stimulation Treatments

Currently, brain stimulation treatments have been proposed as a possible therapeutic option to enhance the recovery of refractory symptoms in several disorders [82,83]. Repetitive Transcranial Magnetic Stimulation (rTMS) is a non-invasive brain stimulation approach that acts by modulating specific brain circuits. While high-frequency (>5 Hz) stimulation determines a depolarization of nerve cells, with long-term potentiation (LTP) effects, low-frequency stimulation protocols (1 Hz) determine the long-term depression (LTD) of the targeted area, with the possibility to induce the localized inhibition of specific disordered networks. According to the cortical hyperexcitability hypothesis about its pathogenesis, several case reports propose that rTMS could be a promising therapeutic method for refractory visual hallucinations in schizophrenia [84,85].

To date, no studies have investigated the potential use of rTMS in HPPD. Interestingly, Kilpatrick and Ermentrout (2012) [86] studied the spatiotemporal dynamics of neuronal networks in HPPD, with spike frequency adaptation. This study reported that altering parameters controlling the strength of synaptic connections in the network can lead to spatially structured activity suggestive of symptoms of HPPD. Future research is necessary to test the possible effectiveness of the rTMS neuromodulatory effect on HPPD. Putative targets of stimulation could be hypothesized to be located in the visual cortical areas, as well as in the occipitotemporal sulcus [87]. Functional neuroimaging may be beneficial in localizing a specific target for stimulation and may prevent wasting time and money on targets which are not as likely to be involved in the pathogenesis.

## 4. Discussion

It has to be highlighted that a limitation of the study might be represented by the search method: in fact, we decided to limit the literature search to the DSM terminology in order to exclude simple “flashback phenomena” that are commonly reported in psychopathology, and that may not follow the use of hallucinogens. This could have narrowed the results, preventing the inclusion of other studies using the ICD terminology, which is less “technical” about the issue.

The main consideration that has to be done with respect to HPPD is its rare and unpredictable nature [16]: current prevalence estimates are unknown, but DSM-5 suggests 4.2% [88]. The condition is more often diagnosed in individuals with a history of previous psychological issues or substance misuse [56], but it can arise in anyone, even after a single exposure (mostly to LSD, but it has also been reported after use of other psychedelics) [89]. In many cases, HPPD may also be explained in terms of a heightened awareness of and concern about ordinary visual phenomena, which is supported by the high rates of anxiety, obsessive-compulsive disorder, hypochondria, and paranoia seen in many patients [90].

The crucial movement towards a comprehensive clinical understanding of Hallucinogen Persisting Perception Spectrum Disorders (HPPSD) [23] is the establishment of an accepted operative nomenclature. This wide spectrum of disorders encompasses different subtypes, ranging from HPPD I to HPPD II, according to our hypothetical distinction. Among the innumerable triggers able to precipitate HPPD, prospectively, the use of natural and synthetic cannabinoids appears to be the most frequent. This is consistent with the rapid and vast diffusion of these novel psychoactive compounds, nowadays easily available without specific cultural filters and references [91,92]. Distinct substances, with completely different mechanisms of action, might lead or precipitate the genesis of HPPD, therefore suggesting a multifaceted etiology. Thus, it is accordingly conceivable that different medications could be useful and helpful in the treatment of different subtypes of HPPD. Tracers and trailing phenomena appear to be the most resistant symptoms. Concomitant coexisting psychiatric disorders can represent a further clinical challenge, with the clinical construct of the lysergic psychoma as a possible heuristic model. According to this theory, the presence of induced psychopathological phenomena (the Psychoma) may trigger a specific reaction excepted by the not-affected part of the mind, trying to counteract the psychoma, which is perceived as a “foreign body in the mind”. Of course, when the psychoma is strong and repeated in its nature, the possibility to determine a full-blown psychosis may become more concrete [93,94].

Regarding treatment options, a combination of medications may be needed according to the preceding or subsequent psychopathology. Given the limited literature about HPPD, a possible hypothesis about the pharmacotherapy of choice in relation to different etiologies has not been considered. However, the presence of psychiatric and neurological comorbidities could represent a valid criterion to address the choice. Clinical experience and an extensive and comprehensive knowledge of these phenomena are vital for successful treatment outcomes.

Controlled clinical investigations are mostly needed in order to better understand the etiology, mechanisms of action, clinical issues, and pharmacological treatment options for Hallucinogen Persisting Perception Spectrum Disorders (HPPSD).

## Figures and Tables

**Figure 1 brainsci-08-00047-f001:**
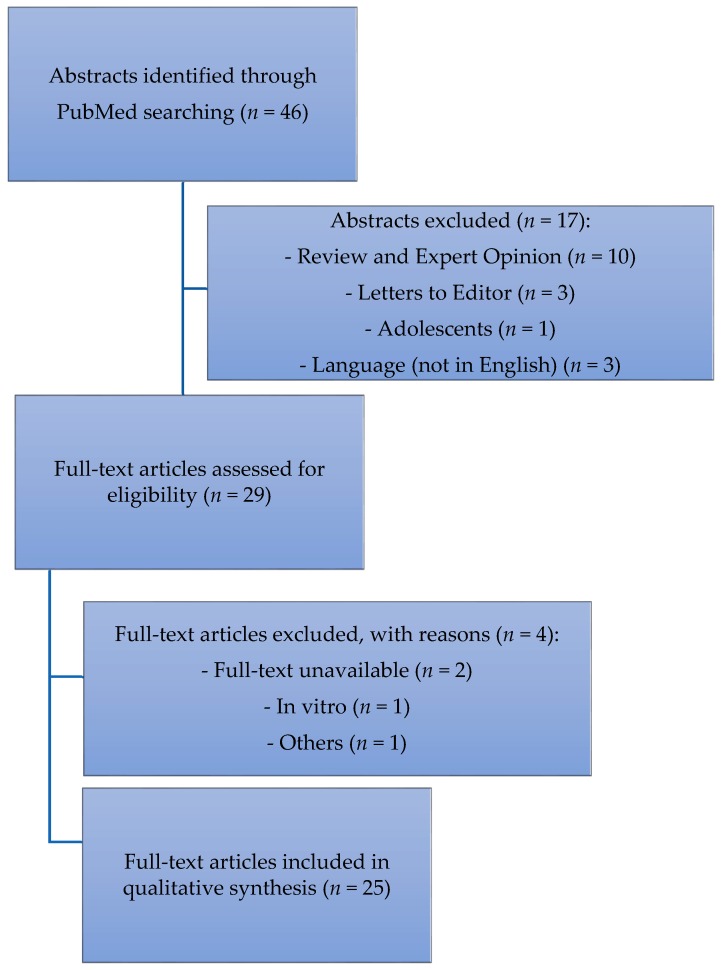
Flow-chart describing the data collection process.

**Table 1 brainsci-08-00047-t001:** Substances that induce Hallucinogen Persisting Perception Disorder (HPPD).

Authors	Cases (*n*)	Substances Inducing Perceptual Disturbances	Trigger Cues
Zobor, 2015 [29]	1	Cannabis	
Gaillard, 2003 [46]	2	Cannabis	
Lerner, 2014 [47]	2	Cannabis (Synthetic)	
Anderson, 2017 [48]	1	Cannabis and MDMA	Stress
Brodrick, 2016 [49]	1	Cannabis and LSD	
Coppola, 2017 [50]	1	Cannabis (Synthetic, JWH-122)	Cannabis consumption
Lerner, 2003 [51]	16	LSD	
Lerner, 2002 [20]	1	LSD	
Lerner, 2000 [52]	8	LSD	
Gaillard, 2003 [46]	1	LSD	Alcohol intake
Lev-Ran, 2017 [53]	40	LSD	Sexual intercourse or Intentional
Hermle, 2012 [54]	1	LSD	Stress
Lerner, 2014 [19]	2	LSD	
Abraham, 2001 [35]	38	LSD	Dark environment
Litjens, 2014 [26]	31	LSD	
Lerner, 2015 [55]	1	LSD	
Baggott, 2011 [56]	104	LSD	
Lev-Ran, 2015 [57]	37	LSD	
Lev-Ran, 2014 [58]	12	LSD	Situation and mental states
Lerner, 1997 [59]	2	LSD	
Abraham, 1996 [34]	3	LSD	
Espiard, 2005 [21]	1	PCP	Cannabis consumption
Lauterbach, 2000 [60]	1	Risperidone	

MDMA: 3,4-methylenedioxy-*N*-methylamphetamine; LSD: lysergic acid diethylamide; JWH-122: 4-methyl-1-(naphthalenyl)(1-pentyl-1H-indol-3-yl)-methanone.

**Table 2 brainsci-08-00047-t002:** A representative, but not exhaustive, list of reported visual disturbances.

Symptom	Description
**Symptom Reported by Diagnostic and Statistical Manual of Mental Disorder, fifth edition (DSM-5)**	
Visual hallucinations	Perceptions in the absence of the objects. False perceived objects are often geometric figures.
Altered motion perception	False perceptions of movement in the peripheral visual fields
Flashes of color	
Color enhancement	Perception of intensified colors
Trails or tracers	Lines, stripes or bands that could be observed after animate and inanimate objects have already moved from their previous location. According to DSM-5, images left suspended in the path of a moving object as seen in stroboscopic photography
Palinopsia	Positive afterimages that continue to appear in one’s vision after the exposure to the original image has ceased.
Halos	Colored light around a light source or an object
Micropsia	Misperception of images as too small
Macropsia	Misperception of images as too large
**Common Symptoms Not Reported by DSM-5**	
Floaters	Spots that seem to drift in front of the eye
Visualizations	Dots, points, particles, mottles or specks emerging in an obscure room
Fractals	Self-similarity perception or small parts that are seen having an equal and identical shape or form as the whole
Repetitions	Recurrence of inanimate or moving patterns or motives
Keenness	Undimmed color contrasts
Pareidolia	An image within an image like the imagery of objects or faces in a foggy arrangement
Superimpositions	Superimposed or overlapped geometric patterns
Distorted Perception of Distance	Objects were seen slightly closer or distant
Monochromatic Vision	The visual perception of distinct colors as one unique color with different tinges and tonalities
Intense fragmentation	The sense of disintegration of still or moving objects
Recurrent Synesthesia	Stimulation of one sensory pathway leads to automatic, involuntary reactions or experiences in a second sensory pathway
Geometric Phosphenes	Seeing light without light penetrating the eye.
Imagistic Phosphenes	Casual and unplanned formed images like non-humans (zoopsia) and human faces without geometric patterns or figures provoked by closing an eye and pressing it with a finger
Acquired Dyslexia	Difficulty with reading notwithstanding normal intelligence
Aeropsia or Visual Snow	Virtually seeing particles of air

**Table 3 brainsci-08-00047-t003:** Observational studies and case reports comparing schizophrenic patients with HPPD (SCZ+HPPD) and schizophrenic patients without HPPD (SCZ) (* *p* > 0.05, ** *p* < 0.05).

Authors	Study	Number of Patients	Substances	Symptoms Description	Onset Perceptual Disorders	Recurrence of Perceptual Disorders	Treatment
Lev-Ran, 2015 [57]	Observational, cross-sectional, control study	80 hospitalized SCZ patient with past use of LSD 43 SCZ (DSM-IV-TR) 37 SCZ+HPPD (DSM-IV-TR) Onset of illness: 22.9 SCZ, 23.4 SCZ+HPPD *	Cannabis: 100% SCZ, 92% SCZ+HPPD * MDMA: 60% SCZ, 46% SCZ+HPPD * Opioids: 26% SCZ, 30% SCZ+HPPD 30% * Cocaine: 16% SCZ, 14% SCZ+HPPD * LSD initiation use: SCZ 17.9y, SCZ+HPPD 19.3y *	Adversive LSD experience (bad trip): 28% SCZ, 89% SCZ+HPPD ** PANSS: Positive symptoms: SCZ = SCZ+HPPD ** Negative symptoms: SCZ > SCZ+HPPD ** General psychopathology: SCZ > SCZ+HPPD ** Total score: SCZ > SCZ+HPPD **		Treatment ineffective in SCZ+HPPD	Antipsychotic medication
Lev-Ran, 2014 [58]	Observational	26 patients 14 SCZ (DSM-IV-TR)12 SCZ+HPPD (DSM-IV-TR) Demographic characteristic did not differ between the two groups	Past use of :LSD (100%) cannabis (100%) MDMA (7%) No differences between the two groups in age at onset of drug use and in number of incidences of hallucinogen use	67% SCZ+HPPD could distinguish HPPD symptoms from hallucination related to a psychotic state	9 SCZ+HPPD patients recognized precursory cues for perceptual distortion (7 substance-induced, 5 situational, and 2 mental cues)	12 patients experienced perceptual distortion (SCZ+HPPD)	Antipsychotic treatment. No significant differences in response to APS and adverse effects between the two groups
Lauterbach, 2000 [60]	Case report	1 psychotic patient	No reported substance abuse and hallucinogen exposure Risperidone Clonazepam Trazodone	HPPD-like symptoms: palinopsia, illusions, and visual disturbances	After risperidone treatment	Weekly recurrence. Remission in 48 h each time	

**Table 4 brainsci-08-00047-t004:** Observational studies and case reports evaluating clinical presentation.

Authors	Study	Number of Patients	Substances	Symptoms Description	Onset Perceptual Disorders	Recurrence of Perceptual Disorders	Treatment
Lev-Ran, 2017 [53]	Observational cross-sectional study	40 (27 males); HPPD (DSM-IV-TR)	Previous use of LSD; Lifetime use of Cannabis	HPPD I: mean age 25.5 (3.7), times of LSD consumption: 7.1 (4.3), use of alcohol; perceptual disorders triggered by sexual intercourse, dark environment, and looking at still or moving objects			None of the subjects included in the study received medications particularly targeted at treating HPPD
				HPPD II: mean age 22.1 (2.8), times of LSD consumption 24.6 (1.4), use of SCS, stimulants and inhalants; intentionally triggering perceptual disturbances			
Zobor, 2015 [29]	Observational, cross-sectional, control study	Male, 23-year-old	Cannabis, previous 4-year history of heavy consumption (16–20 years)	Visual distortion: visual snow, sperm-like whizzing dot, jittering lights, floaters, photophobia, visual discomfort, positive and negative afterimages, impaired night vision, halos, starburst around lights;	During cannabis use period	Persistence despite cannabis withdrawal	No
				Ophthalmological examination: reduction of phosphene threshold, alteration in the EOG			
		4 healthy subjects, mean age 25.5 years	Cannabis: Heavy consumption	Not reported		Not reported	No
Lerner, 2014 [19]	Case report	Male, 24-year-old	Cannabis: Three-year past history of social consumption;	Visual disturbances (halos, color intensification, flashes of colors, distorted perception of distance)	During LSD intoxication	Recurrence one week after completely stopping all substance use: daily visual distortion	Not accepted by the patients
			MDMA, LSD and cocaine (sporadically);				
			Social Alcohol drinking			Disappearance after one year	
		Female, 25-year-old	Cannabis: Three-year past history of social consumption;	Visual disturbances (positive afterimages, color intensification, flashes of colors, trailing phenomena)	During LSD intoxication	Recurrence four days after completely stopping all substance use: daily visual distortion	Not accepted by the patients
			MDMA, LSD (sporadically);			Improvement after one year;	
			Social Alcohol drinking			Trailing phenomena continued to appear intermittently	
Gaillard, 2003 [46]	Case reports	Female, 18-year-old	Cannabis: Three-year past history of regular consumption	White dots when looking at a white wall or blue sky, “seeing shadows” on the left side, palinopsia, visual vibration upon awakening	During comatose episode following excessive use of cannabis	Recurrence after stopping all substance use: daily visual distortion	
		Male, 25-year-old	Cannabis: Two-year past history of regular and heavy consumption	Visual illusion and dyskinetopsia, difficult in depth perception	After two years of consumption	Symptoms persistence and increase after cannabis withdrawal + memory loss, and concentration deficits	
Abraham, 2001 [35]	Observational	38 HPPD cases	LSD: first mean use 18.1 (6.0) years; lifetime use 16 times (median)	7.11 (2.2) different types of visual hallucinations per subject	21 months after first use	Duration of visual hallucinations: 9.67 (7.68) years	
					13.5% subjects experienced symptoms within the first month of use, three subjects after a single use		
				The majority of subjects reported an intensification of visual hallucinations on emerging into a dark environment			
Litjens, 2014 [26]	Case series	31 HPPD cases; Web-questionnaire	MDMA	At least 2 different visual phenomena (visual snow, afterimages, flashes, illusory movement, and increased observation of floaters) with a minimum of one episode of disturbed perception every week (100%);	After a single drug exposure		
			Cannabis				
			LSD	Anxiety or panic in the weeks before or following the use of drugs (71%)			
				Depersonalization (32%)			
		assessment	80% serotonergic drugs	Derealization (39%)	After a period of extensive drug use		
Lerner, 2015 [55]	Case report	Male, 26-year-old	Cannabis: a Five-year history of occasional consumption;	No distressing macropsia, micropsia, pelopsia and teleopsia, looking at still or moving objects and humans;	LSD intoxication	Recurrence two days after completely stopping all substance use: daily visual distortion	Not accepted by the patients
			Alcohol: Social Consumption;				
			LSD: Recreational use	Longer and distressing visual distortion experience with anxiety			
						Disappearance after one year	
Baggott, 2011 [56]	Observational Web-based questionnaire	2679 subjects	Median of 5 different drugs used by a single subject	224 subjects reported having at least one diagnosis associated with unusual visual experiences;	After exposure to LSD	The probability of experiencing constant or near-constant symptoms was predicted by greater past exposure to drugs and exposure to LSD	104 individuals considered symptoms impairing enough to seek treatment
		89.5% male, aged 21.6 (3.7) years		1487 individuals reported at least one abnormal visual experience;			
				587 endorsed at least one experience on a constant or near-constant basis			

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
