# Peer review of "Hallucinogen Persisting Perception Disorder: Etiology, Clinical Features, and Therapeutic Perspectives"

_brainsci, 2018, doi:10.3390/brainsci8030047_

Round 1

Reviewer 1 Report

This is a manuscript that reports on a systematic review of the literature on HPPD, a relatively rare diagnosis made in individuals experiencing perceptual disturbances similar to those experienced while using psychedelics previously. The authors have done a nice job of summarizing the literature, and provides a fairly comprehensive view of the available literature on this little-understood phenomena. However, the weakness of this paper is that the authors largely present a laundry list of the published accounts, without providing much synthesis of the data. For example, despite large amount of reports about specific pharmacologic approaches to treatment, the authors make no attempt to synthesize this in their section describing the possible etiologies. Nevertheless, this might be understandable given the limited literature, the inconsistent nomenclature, and the generally low prevalence of this illness. The style of the writing is also somewhat unconventional, and as such is somewhat difficult to read for an English-speaker. I have included additional minor comments below.

Lines 98-117: The authors speculate on a variety of possible etiologies, but fail to connect to possible etiologies raised in the Treatment section. Authors argue, for example, that alpha-2 agonists, benzos, anti-convulsants, SSRIs, naltrexone, etc may be useful and speculate on the involved neurotransmitter systems.  As such, it might be useful to expand further in this section to include those hypotheses.

line100: The beginning of the sentence is cut off

P6: table 2: Symptom and Description don’t line up well, and very difficult to read

Line 250 and Line 257: Awkward to say “…in reason of efficacy….”  Would re-word to something different.

Line 254: Remove “it” after “Today phenytoin….”

Author Response

This is a manuscript that reports on a systematic review of the literature on HPPD, a relatively rare diagnosis made in individuals experiencing perceptual disturbances similar to those experienced while using psychedelics previously. The authors have done a nice job of summarizing the literature, and provides a fairly comprehensive view of the available literature on this little-understood phenomena. However, the weakness of this paper is that the authors largely present a laundry list of the published accounts, without providing much synthesis of the data. For example, despite large amount of reports about specific pharmacologic approaches to treatment, the authors make no attempt to synthesize this in their section describing the possible etiologies. Nevertheless, this might be understandable given the limited literature, the inconsistent nomenclature, and the generally low prevalence of this illness. The style of the writing is also somewhat unconventional, and as such is somewhat difficult to read for an English-speaker. I have included additional minor comments below.

As correctly noted by the reviewer, the lack of synthesis is a limitation due to the quantitatively poor literature about HPPD. In order to prevent the presentation of speculative hypothesis, we decided for a cautious approach. In agreement with the reviewer, we emphasized this limitation and added some synthetic comments in the discussion session.

Lines 98-117: The authors speculate on a variety of possible etiologies, but fail to connect to possible etiologies raised in the Treatment section. Authors argue, for example, that alpha-2 agonists, benzos, anti-convulsants, SSRIs, naltrexone, etc may be useful and speculate on the involved neurotransmitter systems.  As such, it might be useful to expand further in this section to include those hypotheses.

We have now added a statement in section 3.1, trying to propose a link between etiologies and pharmacotherapies. However, we prefer to take a cautious approach in order to prevent speculations not based on clear evidence.

line100: The beginning of the sentence is cut off

This has been fixed with the manuscript formatting by the editorial office.

P6: table 2: Symptom and Description don’t line up well, and very difficult to read

The table has been modified to make it easier to be read.

Line 250 and Line 257: Awkward to say “…in reason of efficacy….”  Would re-word to something different.

The sentence has been reworded as “because of”.

Line 254: Remove “it” after “Today phenytoin….”

We corrected the typo.

Reviewer 2 Report

The manuscript focuses Hallucinogen Persisting Perception Disorder (HPPD), which is an uncommon, but still relevant condition in patients using specific psychoactive substances. Albeit phenomena which are referred to as HPPD have been known for many decades, to date there are many open questions. The manuscript gives a good overview about several issues associated with HPPD, but there are some important points which should be addressed prior to publication.

- The introduction gives a lot of different information about the use of hallucinogens in past and present. In line 30 and below, the rather unspecific term "hallucinogen" is used without explicitly explaining which groups of substances are included. Do authors refer to "serotonergic hallucinogens" or any kind of hallucinogens (including atypical ones like salvia, etc.)? Are dissociates, entactogens, other stimulants and cannabis also included (as all of them could possibly induce hallucinations)? A brief overview about which substances are dealt with needs to be added here. In the title of the manuscript, the term "psychedelic" is used - could this be a proper word for the substances that are focused in the manuscript?  

- In line 31, the first two characteristics which are mentioned include "entactogenic" and "empathogenic". Both are more associated with the substance group "entactogens" or "empathogens" than with "hallucinogens". In addition, it seems to be over-exact to name both phenomena, which are so closely connected to each other. Instead, other effects more specific for  hallucinogens should be named.

- In the following lines, much is said about the usage and history of hallucinogens (in fact, here again referring mostly to serotonergic hallucinogens and not to the other groups). However, there is a lot of information, with only few references given. I suggest to shorten these parts of the introduction or add more references.

- The treatment of cluster headache with psychedelics originated in the 1990s, not during the 1960s. In the 60s, unspecific forms of headache were "treated" with LSD; treatment success was explained from a psychodynamic viewpoint suggesting a psychosomatic aetiology of headache. In contrast, the treatment of cluster headache was associated with serotonergic effects of psychedelics, emphasising biological underpinnings.

- line 46: In addition to rave parties and social events, there are other contexts for using psychedelics, namely shamanic ceremonies, workshops of underground therapy and self-experiences, and others. It appears to be too narrow not to name these, monovalent use is frequent in these frameworks, whereas heavy polyvalent use is more common in rave parties. This might also have a certain impact on the risks for flashbacks and HPPD.

- line 61-71: it should be mentioned that the distinction between HPPD1 and HPPD2 is of hypothetical nature and that it is suggested by the authors. To date, the underlying neural substrates are not well understood, and we don´t know if both phenomena share similar pathologies. In fact, it should also be discussed how far it makes sense to label a transient phenomenon like flashbacks with a term suggesting persistence. Moreover, this distinction suggests pathologies of different severity where only one ("HPPD2") leads to help seeking behaviour and treatment, whereas "HPPD1" might be experienced as pleasant by many consumers and only in exceptional cases requires treatment. It should also be mentioned that this distinction has not yet been made in DSM-5. 

- line 65: it should also be mentioned that many consumers who experience flashbacks / HPPD1 do not report to suffer from these phenomena; some of them even refer to flashbacks as "free trips", making psychedelic-like experiences in the absence of any substance

- line 105: "explain" instead of explicate?

- line 119: "associated" instead of "correlated"? To my knowledge, no correlations have been found

- line 129: Regarding clinical features, only a "representative" selection of symptoms are made, without naming criteria for selection. It would be helpful to explain in how far are these representative, or to name all symptoms that have been found in the literature search, at least mention all that are constitutive for HPPD in DSM-5. 

- line 130: what does "positive symptoms" mean here? The term evokes symptoms of schizophrenia; are there also negative symptoms? 

- line 140: Frequency in recurrence of symptoms of HPPD1 is lower than for HPPD2. Criteria for a distinction between 1 and 2 remain unmentioned. Is it only the frequency of recurring symptoms which matters? Does that mean, that neither HPPD1 nor HPPD2 is really persisting in a narrow sense of the word? Maybe it would be helpful to add a table with a distinction between both conditions?

- line 204-222: the authors emphasise that benzodiazepines might be helpful in treating benign HPPD1, but not (as) effective in HPPD2. On the other hand, they mention that these medications could be problematic in patients with a history of substance abuse. Finally, it remains unclear why patients with a benign, transient condition like HPPD1 should be exposed to a habit-forming substance and in how far these substances can be suggested as first-line medication, as risks are high and benefits are low?

- line 222-227: in previous research and treatment guidelines, first-generation antipsychotics have been associated with a higher risk of flashbacks / HPPD when administered to patients who are suffering from acute anxiety or psychotic symptoms when under the influence of LSD or other psychedelics. This observation should be addressed.

- line 252-263: Among antiepileptic medications (which appear to be highly promising regarding the different hypotheses on the pathophysiology of HPPD and clinical experiences), lamotrigine is missing, which has been successfully used by Hermle and colleagues (2012, your Ref. 72). Another case report by Anderson (2018) also mentions successful treatment of HPPD with lamotrigine

line 316-320: Again, it should be indicated that the distinction between HPPD1 and HPPD2 is hypothetical nature and not being applied by every author

Author Response

- The introduction gives a lot of different information about the use of hallucinogens in past and present. In line 30 and below, the rather unspecific term "hallucinogen" is used without explicitly explaining which groups of substances are included. Do authors refer to "serotonergic hallucinogens" or any kind of hallucinogens (including atypical ones like salvia, etc.)? Are dissociates, entactogens, other stimulants and cannabis also included (as all of them could possibly induce hallucinations)? A brief overview about which substances are dealt with needs to be added here. In the title of the manuscript, the term "psychedelic" is used - could this be a proper word for the substances that are focused in the manuscript?  

The title has been edited as psychedelics are not the only substances the paper deals with.

In the introduction, it is provided a list of substances that may induce HPPDs (lines 77-82), including also atypical hallucinogens, stimulants such as MDMA, dissociates such as ketamine, and cannabinoids (particularly synthetic cannabinoids). We also added a sentence in order to clarify the association between HPPD and substances other than psychedelics (“It is therefore clear that HPPD is not strictly associated with psychedelic consumption, but a number of hallucinogen-inducing substances may be correlated with its arising.”).

- In line 31, the first two characteristics which are mentioned include "entactogenic" and "empathogenic". Both are more associated with the substance group "entactogens" or "empathogens" than with "hallucinogens". In addition, it seems to be over-exact to name both phenomena, which are so closely connected to each other. Instead, other effects more specific for hallucinogens should be named.

We chose to only leave “empathogenic” as a characteristic of hallucinogens. We also added emotional alterations to the core features of hallucinogens, and a sentence reading “Their main characteristic is to profoundly affect a person’s perception of the surrounding world.”

- In the following lines, much is said about the usage and history of hallucinogens (in fact, here again referring mostly to serotonergic hallucinogens and not to the other groups). However, there is a lot of information, with only few references given. I suggest to shorten these parts of the introduction or add more references.

This section of the introduction has been shortened.

- The treatment of cluster headache with psychedelics originated in the 1990s, not during the 1960s. In the 60s, unspecific forms of headache were "treated" with LSD; treatment success was explained from a psychodynamic viewpoint suggesting a psychosomatic aetiology of headache. In contrast, the treatment of cluster headache was associated with serotonergic effects of psychedelics, emphasising biological underpinnings.

The sentence referring to cluster headache has been erased from the manuscript.

- line 46: In addition to rave parties and social events, there are other contexts for using psychedelics, namely shamanic ceremonies, workshops of underground therapy and self-experiences, and others. It appears to be too narrow not to name these, monovalent use is frequent in these frameworks, whereas heavy polyvalent use is more common in rave parties. This might also have a certain impact on the risks for flashbacks and HPPD.

Thanks to reviewer for making this point. We added this concept to the introduction.

- line 61-71: it should be mentioned that the distinction between HPPD1 and HPPD2 is of hypothetical nature and that it is suggested by the authors. To date, the underlying neural substrates are not well understood, and we don´t know if both phenomena share similar pathologies. In fact, it should also be discussed how far it makes sense to label a transient phenomenon like flashbacks with a term suggesting persistence. Moreover, this distinction suggests pathologies of different severity where only one ("HPPD2") leads to help seeking behaviour and treatment, whereas "HPPD1" might be experienced as pleasant by many consumers and only in exceptional cases requires treatment. It should also be mentioned that this distinction has not yet been made in DSM-5. 

We specified the uncertainty of the distinction between HPPD I and II in the introduction.

- line 65: it should also be mentioned that many consumers who experience flashbacks / HPPD1 do not report to suffer from these phenomena; some of them even refer to flashbacks as "free trips", making psychedelic-like experiences in the absence of any substance

This has been added.

- line 105: "explain" instead of explicate?

The word has been corrected.

- line 119: "associated" instead of "correlated"? To my knowledge, no correlations have been found

Thanks to the reviewer for suggesting a more appropriate term. We modified the phrase accordingly.

- line 129: Regarding clinical features, only a "representative" selection of symptoms are made, without naming criteria for selection. It would be helpful to explain in how far are these representative, or to name all symptoms that have been found in the literature search, at least mention all that are constitutive for HPPD in DSM-5. 

We added further explanation on the symptoms in paragraph 3.3 and in table 2.

- line 130: what does "positive symptoms" mean here? The term evokes symptoms of schizophrenia; are there also negative symptoms? 

We have changed the term “positive” into “perceptive disturbances”. We thank the reviewer for bringing this unclear point to our attention.

- line 140: Frequency in recurrence of symptoms of HPPD1 is lower than for HPPD2. Criteria for a distinction between 1 and 2 remain unmentioned. Is it only the frequency of recurring symptoms which matters? Does that mean, that neither HPPD1 nor HPPD2 is really persisting in a narrow sense of the word? Maybe it would be helpful to add a table with a distinction between both conditions?

To the best of our knowledge, a clear distinction between HPPD I or II is not supported by scientific literature. The possibility to report a table has been ruled out given the speculative nature of this. However, in accordance with the reviewer, we have now emphasized these aspects in the text.

- line 204-222: the authors emphasise that benzodiazepines might be helpful in treating benign HPPD1, but not (as) effective in HPPD2. On the other hand, they mention that these medications could be problematic in patients with a history of substance abuse. Finally, it remains unclear why patients with a benign, transient condition like HPPD1 should be exposed to a habit-forming substance and in how far these substances can be suggested as first-line medication, as risks are high and benefits are low?

In agreement with the reviewer, we tempered our statement and we have now discussed this point.

- line 222-227: in previous research and treatment guidelines, first-generation antipsychotics have been associated with a higher risk of flashbacks / HPPD when administered to patients who are suffering from acute anxiety or psychotic symptoms when under the influence of LSD or other psychedelics. This observation should be addressed.

Thanks to the reviewer for suggesting this observation.  A note on the exacerbation of flashbacks in the early phases of treatment with haloperidol has been added.

- line 252-263: Among antiepileptic medications (which appear to be highly promising regarding the different hypotheses on the pathophysiology of HPPD and clinical experiences), lamotrigine is missing, which has been successfully used by Hermle and colleagues (2012, your Ref. 72). Another case report by Anderson (2018) also mentions successful treatment of HPPD with lamotrigine.

This relevant study has been added and the role of lamotrigine has been emphasized.

-line 316-320: Again, it should be indicated that the distinction between HPPD1 and HPPD2 is hypothetical nature and not being applied by every author

This has been specified in the text.

Reviewer 3 Report

Martinotti et al. review hallucinogen Persisting Perception Disorder, a poorly understood condition. Overall, the authors do a good job in summarizing available knowledge. The following points should be improved:

1.       The title does not match so good with what the authors present in the abstract and in the manuscript. There is no review of Psychedelics (as the title suggests) and the neurobiological studies reviewed generally focus on the actions of LSD, but not HPPD. This should be improved.

2.       The authors introduce HPPD I and HPPD II, but do not focus much on whether there two subtypes are induced by the same or different substances. This would be interesting.

3.       The author contributions and the funding are not given in the manuscript. Please add.

4.       The symptoms which people with HPPD experience do not seem clearly described to me, only that they are positive in nature (p.5). Also, when reading between the lines, it appears to me that the hallucinations are mainly visual but not auditory in nature. This is very interesting, as in schizophrenia, hallucinations are mainly auditory but not visual. The modality in which these hallucinations occur deserves to be treated in more detail.

5.       The authors should fix the page numbers, as they start counting at one again in the middle of the manuscript (e.g. after p. 7). This should be fixed.

6.       In Table 3, it is not mentioned whether patients from the different studies suffer from HPPD I or II. This should be improved, if possible.

7.       The following sentence appears odd: “Although HPPDs are infrequent [16], they are frequently unrecognized [17; 18].”

Author Response

1.       The title does not match so good with what the authors present in the abstract and in the manuscript. There is no review of Psychedelics (as the title suggests) and the neurobiological studies reviewed generally focus on the actions of LSD, but not HPPD. This should be improved.

Thanks to the reviewer for suggesting this improvement. We modified the title of the paper to: “Hallucinogen Persisting Perception Disorder: etiology, clinical features, and therapeutic perspectives”, in order to better match the content of our work.

2.       The authors introduce HPPD I and HPPD II, but do not focus much on whether there two subtypes are induced by the same or different substances. This would be interesting.

All of the substances mentioned in the introduction have been associated to HPPD phenomena, but so far, no further distinction on substances inducing HPPD I or II is currently supported by scientific literature.

3.       The author contributions and the funding are not given in the manuscript. Please add.

We did not receive any funding for the work reported in this paper, and we added a statement with respect to this. The journal guidelines requests to specify the contribution of each author for research papers, and not for reviews: this is why we did not add a paragraph on this matter.

4.       The symptoms which people with HPPD experience do not seem clearly described to me, only that they are positive in nature (p.5). Also, when reading between the lines, it appears to me that the hallucinations are mainly visual but not auditory in nature. This is very interesting, as in schizophrenia, hallucinations are mainly auditory but not visual. The modality in which these hallucinations occur deserves to be treated in more detail.

We added further details on the nature of hallucinatory symptoms in paragraph 3.3, and a brief note in the introduction, citing a study by Caton et al, which highlights that in the vast majority of induced psychoses, visual hallucinations are notably more common than auditory.

5.       The authors should fix the page numbers, as they start counting at one again in the middle of the manuscript (e.g. after p. 7). This should be fixed.

This has been fixed by the manuscript formatting of the journal’s editorial office.

6.       In Table 3, it is not mentioned whether patients from the different studies suffer from HPPD I or II. This should be improved, if possible.

The authors of the papers mentioned in Table 3 did not differentiate patients suffering from HPPD I or II, probably because this is distinction is not included in DSM-5, and is therefore “unofficial”. Moreover, it has to be considered that the studies were related to schizophrenic patients, which makes it extremely difficult to discriminate between the HPPDs and endogenous psychotic symptoms.

7.       The following sentence appears odd: “Although HPPDs are infrequent [16], they are frequently unrecognized [17; 18].”

The sentence has been rewritten as “HPPDs do not have a notable prevalence[16], and, therefore, they are frequently unrecognized [17; 18].”

Round 2

Reviewer 2 Report

A sentence has been added mentioning that the "main characteristic (of psychedelics) is to profoundly affect a person’s perception of the surrounding world.” It is correct, that these substances might change the perception of the surrounding world, but this phenomenon is only describing one particular aspect of the effects, as the perception of inner processes is also affected. in other words, it´s not only the perception of surrounding world, but also profound changes of introspective aspects affected by these substances.

Author Response

We would like to thank the reviewer for the comment. We modified the sentence, that now reads:  "Their main characteristic is to profoundly affect a person’s inner processes, and the perception of the surrounding world". 

Reviewer 3 Report

All my comments have been adequately addressed.

Author Response

We would like to thank the reviewer for giving us the chance to significantly improve our work.